# Serum 25-Hydroxyvitamin D Levels and Youth-Onset Type 2 Diabetes: A Two-Sample Mendelian Randomization Study

**DOI:** 10.3390/nu15041016

**Published:** 2023-02-17

**Authors:** Benjamin De La Barrera, Despoina Manousaki

**Affiliations:** 1Research Center of the Sainte-Justine University Hospital, University of Montreal, Montreal, QC H3T 1C5, Canada; 2Departments of Pediatrics, Biochemistry and Molecular Medicine, University of Montreal, Montreal, QC H3T 1C5, Canada

**Keywords:** vitamin D, pediatric type 2 diabetes, Mendelian randomization, GWAS, causal inference

## Abstract

Observational studies have linked vitamin D insufficiency to pediatric type 2 diabetes (T2D), but evidence from vitamin D supplementation trials is sparse. Given the rising prevalence of pediatric T2D in all ethnicities, determining the protective role of vitamin D has significant public health importance. We tested whether serum 25-hydroxyvitamin D (25OHD) levels are causally linked to youth-onset T2D risk using Mendelian randomization (MR). We selected 54 single-nucleotide polymorphisms (SNPs) associated with 25OHD in a European genome-wide association study (GWAS) on 443,734 individuals and obtained their effects on pediatric T2D from the multi-ethnic PRODIGY GWAS (3006 cases/6061 controls). We applied inverse variance weighted (IVW) MR and a series of MR methods to control for pleiotropy. We undertook sensitivity analyses in ethnic sub-cohorts of PRODIGY, using SNPs in core vitamin D genes or ancestry-informed 25OHD SNPs. Multivariable MR accounted for the mediating effects of body mass index. We found that a standard deviation increase in 25OHD in the logarithmic scale did not affect youth-onset T2D risk (IVW MR odds ratio (OR) = 1.04, 95% CI = 0.96–1.13, *p* = 0.35) in the multi-ethnic analysis, and sensitivity, ancestry-specific and multivariable MR analyses showed consistent results. Our study had limited power to detect small/moderate effects of 25OHD (OR of pediatric T2D < 1.39 to 2.1). In conclusion, 25OHD levels are unlikely to have significant effects on the risk of youth-onset T2D across different ethnicities.

## 1. Introduction

Type 2 diabetes (T2D) is considered a global epidemic in both adults and youth [1]. Previously a rare disease in children, T2D is becoming increasingly prevalent in children and adolescents worldwide and in all ethnicities, even if the prevalence of obesity remains stable [2]. Notably, T2D accounts for up to 45% of cases of diabetes in youth in certain at-risk populations [3]. Furthermore, individuals who develop T2D in childhood and adolescence have a substantial risk of T2D-related complications at a young age [4]. Thus, the prevention of youth-onset T2D presents a particularly important yet difficult challenge. Primary prevention through lifestyle-based approaches aiming to reduce the burden of obesity is paramount, but the long-term success of such interventions is limited [5]. Therefore, identifying novel biomarkers as modifiable risk factors or treatable targets for the primary prevention of T2D in youth is a public health priority. 

Due to its potential immune-modulating and anti-inflammatory properties, vitamin D has been extensively studied as a biomarker for the autoimmune type 1 diabetes [6], but data on its role on youth-onset T2D are sparse. Case-control studies have demonstrated that adolescents with insulin resistance display lower levels of 25 hydroxyvitamin D (25OHD), the biomarker of vitamin D in humans [7], but this does not necessarily apply to pediatric patients with prediabetes or T2D [8,9]. Various studies suggest that vitamin D supplementation can reverse the progression of prediabetes, a condition present in up to 20% of adolescents, to overt type 2 diabetes [10,11]. Many large randomized controlled trials have investigated a potential effect of vitamin D in adult T2D [12], with mostly negative results. In the D2d trial [11], a daily dose of 4000 units of vitamin D showed a non-significant trend to delay the progression of prediabetes into T2D, but in post hoc analysis, a significant effect was found in overweight individuals with severe vitamin D deficiency at baseline, who displayed an increase in serum 25OHD. In a meta-analysis combining the results of the D2d trial with two other trials, vitamin D supplementation reduced the risk of developing T2D by up to 13% in participants who had prediabetes but were not necessarily vitamin D deficient [13]. The available evidence on the benefit of vitamin D supplementation in preventing T2D in adolescents with insulin resistance is contradictory [14,15]. A meta-analysis of vitamin D randomized controlled trials (RCTs) in obese adolescents has shown an improvement in HOMA-IR, a marker of insulin resistance [16]. A recent study from Iran demonstrated the cost-effectiveness of population-wide supplementation with 50,000 units of vitamin D monthly in youth to prevent T2D in adulthood [17]. Against this backdrop, better evidence is needed to understand if vitamin D insufficiency, which affects up to 40% of children and adolescents among different ethnicities [18,19,20], predisposes youth to early-onset T2D.

One of the approaches which has been widely used to study causality between vitamin D and T2D in adults [12] is Mendelian randomization (MR). Under specific assumptions, MR uses single-nucleotide polymorphisms (SNPs) as instruments for a modifiable exposure to study the causal effects of this exposure on a disease outcome [21]. The effects of these SNPs on both exposure and outcome are derived from large genome-wide association studies (GWAS). Due to the random allocation of genetic variants at conception, MR allows for causal inference by limiting bias caused by confounding and reverse causation hampering observational studies, since these biases cannot affect the germline genetic architecture of an individual. As such, MR provides evidence of the effects of lifetime exposure to a genetically determined level of an exposure on a disease outcome.

In this study, we aimed to test whether genetically altered 25OHD levels are causally associated with the risk of youth-onset T2D in mixed-ancestry and ethnic-specific cohorts using MR. To do this, we leveraged data from the largest available European and ancestry-specific GWAS on 25OHD levels [22,23] and on the only available multi-ethnic GWAS on pediatric T2D [24].

## 2. Materials and Methods

Our methods and findings are reported according to the MR-STROBE checklist (Appendix A).

### 2.1. SNPs Associated with 25OHD Levels

In order to test the causality of 25OHD on the risk of pediatric T2D within the MR framework, we first obtained conditionally independent SNPs associated with 25OHD in a European GWAS meta-analysis of the SUNLIGHT consortium with UK Biobank totaling 443,734 individuals [22]. In this GWAS, the mean 25OHD level, measured using the Diasorin assay, was 70 nmol/L (SD 34.7 nmol/L). Levels of 25OHD were adjusted for age, sex, season of 25OHD measurement in the entire GWAS and additionally for vitamin D supplementation in the UK Biobank subset of the GWAS meta-analysis. The effects of these SNPs on the risk of pediatric T2D were sought in the multi-ethnic PRODIGY GWAS cohort (*n* = 3006 youth cases [mean age 15.1 years] and 6061 adult controls) [24]. The 3006 cases of the PRODIGY combined 449 youth with T2D from the TODAY study with >2000 adolescents with T2D from a TODAY ancillary genetics study, and 468 adolescents with T2D from SEARCH for Diabetes in Youth. The adult controls were retrieved from the T2D-GENES study.

We also undertook ethnic-specific analyses by extracting effects of the 25OHD SNPs in the three ethnic sub-cohorts of PRODIGY (Non-Hispanic Whites, *n* = 664 cases/1434 controls; African Americans, *n* = 1068 cases/1068 controls; and Hispanics, *n* = 1274 cases/3559 controls). Descriptions of the GWAS populations can be found in the respective GWAS publications [22,24]. For 25OHD-related SNPs not directly found in the ethnic-specific PRODIGY GWAS, we sought proxy SNPs (LD *r*^2^ > 0.7) using ldlink [25] and its LDproxy function in matching populations from the 1000 genomes phase 3 panel. For our main MR analysis, we computed the Wald ratios of the SNP-IVs and meta-analyzed these individual Wald rations using the inverse-variance weighted (IVW) approach [26]. The results of our MR analyses are expressed as the odds ratio (OR) of pediatric T2D associated with one standard deviation (SD) increase in the log-transformed level of 25OHD (which represents a 40.9 nmol/L change in serum 25OHD level in a vitamin D sufficient individual). MR *p*-values below 0.05 were considered significant. Figure 1 illustrates the flowchart of our analyses and Figure 2 depicts the MR direct acyclic graph of our study.

### 2.2. MR Assumptions

In order to conduct any MR study, the variants used as instrumental variables (IVs) of an exposure must satisfy three assumptions. The first assumption (relevance assumption) requires that these SNPs should be strongly associated with the exposure; in this case, the 25OHD levels. This is ensured by using SNPs linked to 25OHD at a genome-wide significance level (*p*-value < 5 × 10^−8^). We also calculated the F-statistic for the 25OHD SNP-IVs, as an additional measure of the strength of our MR instruments.

The second assumption (independence assumption) requires that the SNPs used as IVs should not be linked to confounders of the association between the exposure and outcome. For instance, in this study, a possible confounder may be the body mass index (BMI), since obesity is associated with lower 25OHD levels [27] and truncal adiposity is an established risk factor for both adult and pediatric T2D [1]. Another confounder of the association between 25OHD and T2D risk is ethnicity. For instance, African Americans present lower 25OHD levels and increased risk of T2D [28]. We undertook multiple sensitivity analyses with different sets of 25OHD SNPs to ensure that our MR results are not biased by the above confounders. Additionally, we performed a multivariable MR analysis considering both 25OHD and pediatric BMI as exposures. The third MR assumption (exclusion restriction assumption) requires that the SNP-IVs affect the outcome (here, pediatric T2D) solely via the exposure (here, 25OHD levels). Pleiotropy refers to a situation in which this assumption is violated. In order to test for the presence of pleiotropy, we conducted sensitivity analyses applying various pleiotropy-robust MR methods, each with its own assumptions.

### 2.3. Sensitivity Analyses Addressing Bias Due to Confounding

Using the PhenoScanner database [29], we filtered the MR instruments for 25OHD for SNPs with previously reported GWAS association with confounders of the 25OHD-pediatric T2D association. A detailed description of this approach can be found in previous MR studies by our group [30,31]. Specifically, we conducted sensitivity MR analyses excluding 25OHD SNPs presenting genome-wide associations with BMI, body composition traits and adult-onset T2D, since family history of adult-onset T2D is a strong risk factor for youth-onset T2D.

To further account for effects of BMI on our MR estimates, we conducted a multivariable MR (MVMR) analysis [32]. To do this, we queried the effects of 25OHD SNPs on childhood BMI from a large European GWAS meta-analysis by the EGG consortium (5530 cases/8318 controls) [33].

Finally, to further account for confounding due to ancestry, we conducted a sensitivity analysis selecting three directly matching rare SNPs (rs14355701 in *TINK*, rs116950775 in *KIAA1644/LDOC1L* and rs111955953 in *FTMT*) from an African American 25OHD GWAS on 697 individuals [23], which were combined to a common SNP in *GC* (rs4588) identified in a recent larger GWAS meta-analysis on 2602 African Americans from the Southern Community Cohort Study and 6934 participants with African or Caribbean ancestry from the UK Biobank [34]. Effects (betas) of these SNPs on 25OHD were extracted from the above African 25OHD GWAS and their effects on pediatric T2D were identified in the African American subset of PRODIGY. 

### 2.4. Sensitivity Analyses Addressing Pleiotropy

We employed four pleiotropy-robust MR methods (weighted median, weighted mode, MR-Egger, and MR-PRESSO) to investigate if pleiotropy could have biased the MR estimates of our main IVW MR analysis. Each of the above methods has its own assumptions.

Specifically, the weighted median method can generate valid estimates if more than 50% of the SNPs used as instruments are valid. This method is based on the fact that estimates of SNPs without pleiotropic effects are usually found closer to the median, while those of pleiotropic SNPs are found further from the median and introduce heterogeneity [35]. The weighted mode method is very similar to the weighted median method, with the only difference being that it identifies SNPs without pleiotropic effects as the ones with estimates near the mode [36]. The MR-Egger method [37] detects potentially unbalanced directional pleiotropy. This method generates a regression slope, which signifies the causal estimate, and an intercept, which signifies the presence of directional pleiotropy if it differs from the null. To do this, MR-Egger allows for balanced pleiotropy, but requires that the association of each IV with the exposure is not proportional to its pleiotropic effect (The Instrument Strength Independent of Direct Effect or InSIDE assumption). Moreover, the Mendelian Randomization Pleiotropy RESidual Sum and Outlier (MR-PRESSO) method detects, excludes outlier SNP with potential pleiotropic effects and tests for significant distortion of the main MR estimate after outlier removal (global, outlier and distortion tests) [38]. As an additional approach to detect pleiotropy, we computed heterogeneity estimates for each 25OHD SNP in our IVW and MR-Egger analyses using the Cohran-Q metric. Finally, we applied the Steiger directionality test [39] to investigate if the direction of the association (i.e., the assumption than exposure causes an outcome) was valid in our MR analyses. 

As a further strategy to address bias due to pleiotropy, we undertook MR analyses using four SNPs for 25OHD mapping in genes with a known role in vitamin D metabolism [40] (rs12785878 in *NADSYN1*-*DHCR7*, rs10741657 in *CYP2R1*, rs3755967 in *GC* and rs17216707 in *CYP24A1*) since SNPs in these genes are unlikely to be pleiotropic. The effects (betas) on 25OHD of these four SNPs were derived from the Jiang et al. vitamin D GWAS [41].

The TwoSampleMR R package (version 0.5.6) [42] and its default parameters (LD-clumping *r*^2^ = 0.001) were used to retrieve 25OHD SNPs, harmonize them between the exposure and outcome GWAS and compute the various MR estimates [IVW, weighted median, MR-Egger and weighted mode] in our analyses. Random effects IVW was used, given the presence of heterogeneity in our main analyses. Scatter plots and forest plots of our MR studies were prepared using the TwoSampleMR R package. We used the MVMR R package for our multivariable MR analysis [43]. Our MR-PRESSO analyses were applied using the MR-PRESSO R package (version 1.0) [38].

### 2.5. Statistical Power Analysis

We computed the power in our main MR analyses using the full set of 25OHD SNPs in the mixed-ancestry PRODIGY or the ethnic-specific PRODIGY sub-cohorts using an established power calculation method for MR [44]. Specifically, we computed the MR odds ratio (OR) for pediatric T2D for which our analyses obtained a power of 80%, setting the alpha level at 0.05, using the variance of 25OHD explained by its respective SNP-IVs, and the sample sizes of the entire PRODIGY and its ethnic sub-cohorts.

## 3. Results

### 3.1. Main MR Studies on the Effect of Serum 25OHD on Risk of Pediatric T2D across Different Ancestries

Among 138 independent genome-wide significant SNPs reported in the 25OHD GWAS, we first excluded 4 ambiguous ones (i.e., SNPs with non-concordant alleles, e.g., A/G vs. A/C). We retained 87 SNPs with minor allele frequency (MAF) > 1%, since rare variants were unlikely to be found in the mixed ancestry meta-analysis and the ethnic-specific GWAS of PRODIGY. Following LD clumping (*r*^2^ < 0.001), the SNP number was reduced to 57. In the entire PRODIGY, and its Non-Hispanic White, African American, and Hispanic sub-cohorts, we found 49, 49, 46 and 47 direct matches among the 57 SNPs (Appendix A). We were able to identify five proxy SNPs with LD *r*^2^ > 0.8 in the Non-Hispanic White subset of the PRODIGY GWAS but not in the other MR analyses. In our main MR studies, the 49, 54, 46 and 47 common 25OHD SNPs explained 3.1% of the variance in 25OHD levels in Non-Hispanic Whites and 2.5% of the variance in 25OHD in the mixed-ancestry meta-analysis in African Americans and in Hispanics, respectively. Using these SNPs as instruments, we did not find any significant MR association between 25OHD levels and pediatric T2D risk in the mixed-ancestry PRODIGY cohort and its ethnic-specific subsets (Appendix A, Figure 3, Appendix A). As shown in Appendix A and Figure 3, our MR estimates remained largely consistent and close to the null across different MR methods, with the exception of the weighted median MR demonstrating a marginal effect of an SD increase in log-transformed 25OHD on pediatric T2D risk in the mixed-ancestry cohort (OR pediatric T2D 1.09, 95% CI 1.00–1.18, *p* = 0.049) and in Hispanics (OR 1.13, 95% CI 1.02–1.26, *p* = 0.019). Nevertheless, these results were not supported by the MR-Egger, IVW and the weighted mode methods in the mixed-ancestry cohort and by the IVW and weighted mode methods in the Hispanic sub-cohort. Given the consistency of our results across different methods and ancestries, we conclude that the above positive findings, while suggestive, cannot unequivocally prove that 25OHD plays a role in pediatric T2D.

As shown in Appendix A, the intercept of the MR-Egger showed no evidence of unbalanced horizontal pleiotropy in any of the MR studies, but there was significant heterogeneity among the MR instruments in all MR analyses except the one in African Americans. All 25OHD SNP-IVs in our MR analyses had an F-statistic > 10 (their average F-statistic was 234) (Appendix A). The Steiger directionality test indicated that the direction of the examined causal association in our main MR analyses was valid, and as such reverse causation did not bias the reported MR effects.

### 3.2. Sensitivity MR Analyses

As demonstrated in Appendix A and Table 1, the results of our sensitivity analyses using four SNPs in core vitamin D genes (explaining 1.03% of the variance in 25OHD), as well as the four African American 25OHD SNPs, did not indicate any causal effect of 25OHD on pediatric T2D either. Our sensitivity analysis excluding SNPs associated with BMI, body composition traits or adult T2D in the PhenoScanner database provided similar results to those of the main analysis (Appendix A) with the exception of a significant estimate of the weighted median analyses in the mixed-ancestry and the Hispanic PRODIGY cohorts. Finally, the results of the MVMR testing for mediating effects of pediatric BMI were equally non-significant (Appendix A).

### 3.3. MR Power Calculation

Our main MR study had 80% power to identify effects as small as an OR of 1.39 for pediatric T2D per 1 SD increase in log-transformed 25OHD in the mixed-ancestry PRODIGY cohort. The respective power estimates for the ancestry-specific MR analyses were an OR of 1.66 for Hispanics, 2.00 for Non-Hispanic Whites and 2.10 for African Americans (Appendix A). 

## 4. Discussions

Using an unbiased MR approach, and leveraging data from the largest available GWAS datasets for 25OHD and pediatric T2D across different ancestries, we did not observe evidence of a large causal effect of 25OHD levels on risk of pediatric T2D. Additionally, a plethora of sensitivity analyses addressing pleiotropy and accounting for possible confounding effects of BMI provided null results consistent with those of the main IVW analyses. While our study is not powered to exclude small to moderate effects in certain ethnic groups, we can reasonably conclude that it is unlikely that variation in 25OHD levels within the normal distribution substantially affects an individual’s predisposition to develop T2D early in life.

Our results suggest that the low 25OHD levels found in adolescents with insulin resistance in an observational study [7] are unlikely to be causal for their altered glucose metabolism, but they are rather driven by confounders, such as obesity. On the other hand, adolescents with obesity tend to be less physically active and spend less time outdoors, and as such can be less exposed to sunlight and have decreased vitamin D synthesis in the skin [9]. Moreover, they often have poor dietary habits and suboptimal vitamin D intake [27]. Another possible explanation for the observed associations between low 25OHD levels and risk of pediatric T2D is ancestry. In this respect, a study [8] showed that although vitamin D deficiency or insufficiency appeared to affect a substantial proportion of youth with T2D, notably non-Europeans, their prevalence was comparable to that in youth without diabetes. Taken together, all the above mechanisms highlight the presence of various possible confounders in the association between vitamin D and pediatric T2D. 

The results of our MR study are in line with those of MR studies showing an absence of causal association of 25OHD levels with adult T2D. At least five well-powered MR studies, totaling over 500,000 adult cases with T2D of mostly European ancestry, did not find a causal role of genetically determined 25OHD levels on T2D risk [45,46,47,48,49]. In non-Europeans, an MR study in Chinese [50] showed equally null results for T2D risk, fasting plasma glucose and HbA1c. Interestingly, a more recent Chinese MR study [51] on 2393 participants (361 cases with T2D) showed a marginal protective effect of higher 25OHD levels on T2D risk using two vitamin D synthesis SNPs in *CYP2R1* and *DHCR7*, but no effect when using four SNPs in both synthesis and metabolism genes (*CYP2R1*, *DHCR7*, *GC*, and *CYP24A1*).

Our MR study has a few considerable limitations. First, we did not study causal effects of changes in the active form of vitamin D [1,25 di-hydroxyvitamin D or 1,25(OH)2D] due to lack of availability of GWAS data on this form of vitamin D allowing us to extract IVs for 1,25(OH)2D. Another limitation of our study is that the 25OHD SNPs which were used as instruments explain only up to 3.1% of variance in 25OHD levels in the Non-Hispanic White analysis, and only 2.5% of its variance in the other MR analyses. This, combined with the limited sample size of the PRODIGY GWAS, and in particular of its ancestry-specific sub-GWAS, restricted the power of our study to identify small and moderate effects of 25OHD on pediatric T2D risk. We elected to use 25OHD SNPs identified in a large European vitamin D GWAS as instruments in our main mixed-ancestry and ancestry-specific MR analyses. This could have introduced bias due to the fact that effects of these SNPs on 25OHD levels in non-Europeans can differ substantially from those identified in a European GWAS. Nevertheless, there is no available 25OHD GWAS in Hispanics, and the results of our sensitivity analysis using African American-specific 25OHD SNPs were similar to those of the analysis using European 25OHD SNPs. Another limitation is the fact that both the 25OHD and youth-onset T2D GWAS included participants from restricted geographical areas, and as such the findings need to be validated in ethnically diverse populations with a more varied geographic distribution. Future large ancestry-informed GWAS on 25OHD and pediatric T2D will also enable greater power to interrogate their association in various ethnic populations, including populations not studied in this work, using MR. Finally, the two-sample MR design of our study did not allow us to undertake a stratified MR analysis to assess non-linear effects of 25OHD levels, since there are no available 25OHD data in PRODIGY participants. As such, effects of extremely low or high 25OHD levels on risk of pediatric T2D cannot be excluded. 

## 5. Conclusions

In conclusion, our MR study is the first to interrogate the causal association between 25OHD levels and the risk of T2D in youth across different ancestries. The absence of evidence for a large causal effect of 25OHD on the risk of pediatric T2D can inform decisions on conducting clinical trials or public health interventions. Our results do not support vitamin D supplementation as a measure to prevent T2D in youth of any ethnic background, but we cannot exclude small to moderate causal effects. Targeting lifestyle habits, which affect both 25OHD levels and established risk factors for pediatric T2D, such as obesity, is likely a more promising strategy than using vitamin D supplements to prevent T2D in youth without frank vitamin D deficiency. 

## Figures and Tables

**Figure 1 nutrients-15-01016-f001:**
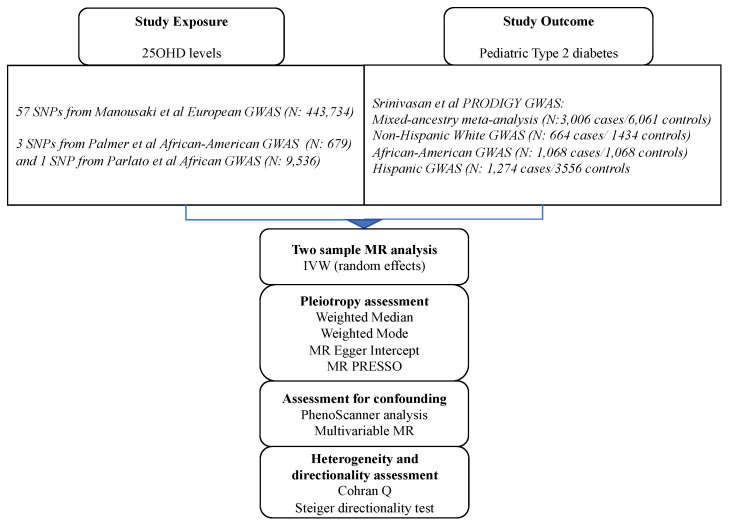
Flowchart with the design of our MR study.

**Figure 2 nutrients-15-01016-f002:**
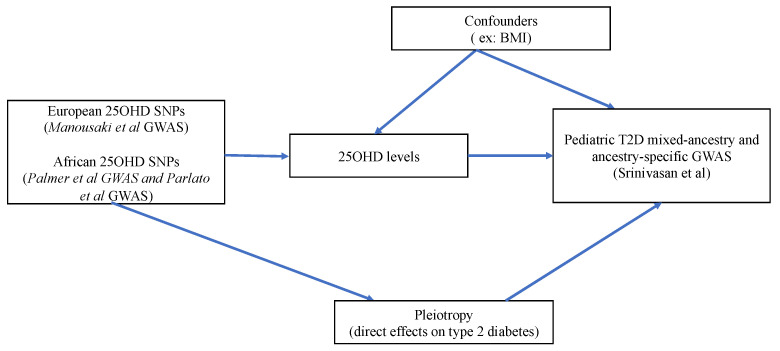
MR Direct Acyclic Graph (DAG) of our study.

**Figure 3 nutrients-15-01016-f003:**
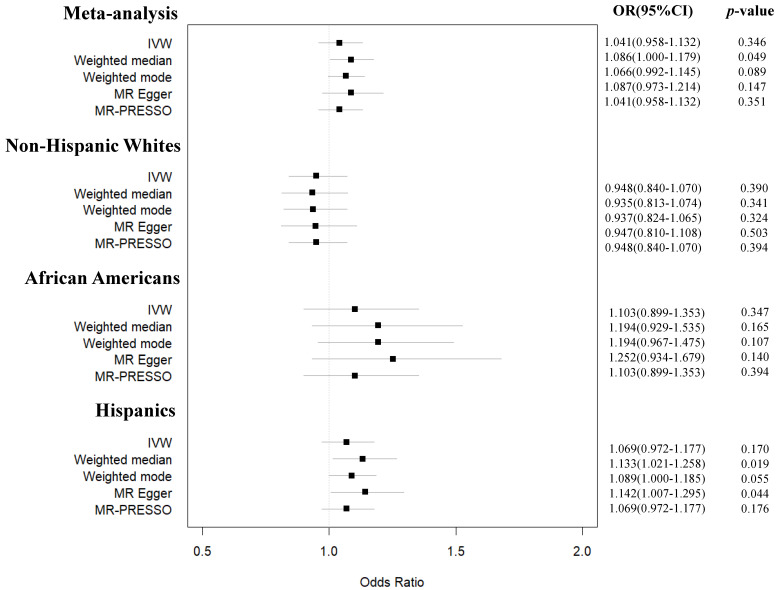
Forest plots with the results (in the form of MR OR for pediatric T2D per SD increase in log-transformed 25OHD) of the main MR analysis in the mixed-ancestry and ancestry-specific PRODIGY cohorts.

**Table 1 nutrients-15-01016-t001:** Results of the MR sensitivity analysis using 4 African 25OHD SNPs.

Method	OR	95%_CI	*p*-Value
IVW	0.991	0.935–1.050	0.763
Weighted median	0.984	0.92–1.0502	0.624
Weighted mode	0.983	0.920–1.050	0.643
MR Egger	1.065	0.904–1.254	0.531

## Data Availability

R scripts used to generate the results of this study are available upon request to the corresponding author. Summary-level results of all GWAS used in this study are publicly available through GWAS catalog.

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
