# Peer review of "Serum 25-Hydroxyvitamin D Levels and Youth-Onset Type 2 Diabetes: A Two-Sample Mendelian Randomization Study"

_nutrients, 2023, doi:10.3390/nu15041016_

Round 1
Reviewer 1 Report
This MR is well strucuterd and well-written. I only have some minor comments:
Please revise the text for typos (e.g. mode instead of model).
How can the total N of SNP for the Entier Cohort being higher than that of the African-American alone (49 vs 46?)
This sentence can be misinterpreted:" The Steiger directionality test indicated that the correct causal direction was “TRUE” in all our main MR analyses, confirming that the assumption that altered 25OHD levels cause pediatric T2D (and not the inverse) was valid."
It might be better to say that the assumption that T2D did not influence vitD levels was valid.
In any case Please clarify in the methods how this was evalaugted, did you extract all SNPs associated with T2D and tested their association with VitD levels (reverse MR?).
Tables S6 are important and should be reported in the main text. Please cite the 4 genes included in this analyses.
Please consider converting or adding some figures as forest plot that help grasping the results directly.
Author Response
We would like to thank the Editor and the reviewers for their constructive critique. Below are our point to point answers to the reviewers’ comments. We have also increased the word count of the current version to 4,021 (including description of Supplemental material, Table and Figure legends, and acknowledgments, but not the references)
We hope that the manuscript in its current version satisfies the criteria for publication in Nutrients.
Reviewer 1
How can the total N of SNP for the Entier Cohort being higher than that of the African-American alone (49 vs 46?).
The reason for having less SNP instruments in the entire cohort compared to the ethnic-specific cohorts is that in the ethnic-specific cohorts we were able to retrieve proxy SNPs based on LD r2 calculated in ethnic-specific 1000 genome populations. We avoided performing a proxy search in the entire cohort, due to fact that proxy search in a mixed-ethnic population is less precise.
This sentence can be misinterpreted:" The Steiger directionality test indicated that the correct causal direction was “TRUE” in all our main MR analyses, confirming that the assumption that altered 25OHD levels cause pediatric T2D (and not the inverse) was valid." It might be better to say that the assumption that T2D did not influence vitD levels was valid. In any case Please clarify in the methods how this was evalaugted, did you extract all SNPs associated with T2D and tested their association with VitD levels (reverse MR?).
We did not perform a reverse MR, but used the Steiger directionality test to evaluate if reverse causation has influenced the results of our forward MR. We rephrased this sentence as follows: “The Steiger directionality test indicated that the direction of the examined causal associations in our main MR analyses was valid, and that reverse causation did not bias the reported MR effects”.
Tables S6 are important and should be reported in the main text. Please cite the 4 genes included in this analyses.
We now include Table S6 in our main manuscript (Table 1). The names of the 4 genes encompassing the 4 SNPs appear in the methods section of the manuscript: “to further account for confounding due to ancestry, we conducted a sensitivity analysis selecting three directly matching rare SNPs (rs14355701 in TINK, rs116950775 in KIAA1644/LDOC1L and rs111955953 in FTMT) from an African-American 25OHD GWAS on 697 individuals[22], which were combined to a common SNP in GC (rs4588) identified in a recent larger GWAS meta-analysis on 2,602 African Americans from the Southern Community Cohort Study and 6,934 African- or Caribbean-ancestry participants from the UK Biobank[33].”
Please consider converting or adding some figures as forest plot that help grasping the results directly.
We have added the results of Table 1 in the form of a forest plot as Figure 3 of the manuscript. To avoid to be redundant, we moved Table 1 to the Supplement, and it now appears as Table S5.
Reviewer 2 Report
In the paper titled: ”Serum 25-Hydroxyvitamin D Levels and Youth-Onset Type 2 Diabetes: A
Two-Sample Mendelian Randomization Study”, the aim was to test whether genetically altered 25OHD levels are causally associated with risk of youth-onset T2D in a mixed-ancestry and in ethnic-specific cohorts using MR.
The paper is interesting, well written and gives comprehensive literature.
In an era of numerous vitamin D studies and growing results revealing its new roles and importance, findings that vitamin D has no significant effect on ”something” is almost brave.
I suggest the following minor corrections:
In the section Materials and Methods (line 76)
1. Since 25OHD levels can be significantly altered due to different exposure to UVB, in a European GWAS meta-analysis, Manousaki et al, selected “winter” individuals assessed January-March (n = 98,674) and “summer” individuals assessed July-September (n = 95,135), while Individuals with vitamin D levels assessed in spring (April-June) and fall (October-December) were not included.
Can you clarify in what season the vitamin D concentration measurements were performed in subjects included in this study?
2. Where there any vitamin D supplement users in your study cohort?
3. If yes, did you performed any adjustments of 25OHD levels?
In the section Discussion (line 269)
Besides limitations which are stated by authors, I would add one more, which refers to the fact that all individuals were from the same or similar geographical area. Even though mixed-ancestry was one of the focus of examination, there is no geographical diversity among them.
Author Response
We thank the reviewer for his constructive critique. Our answers to his/her comments appear below in italics.
In the paper titled: ”Serum 25-Hydroxyvitamin D Levels and Youth-Onset Type 2 Diabetes: A
Two-Sample Mendelian Randomization Study”, the aim was to test whether genetically altered 25OHD levels are causally associated with risk of youth-onset T2D in a mixed-ancestry and in ethnic-specific cohorts using MR.
The paper is interesting, well written and gives comprehensive literature.
In an era of numerous vitamin D studies and growing results revealing its new roles and importance, findings that vitamin D has no significant effect on ”something” is almost brave.
We thank the reviewer for his/her positive feedback.
I suggest the following minor corrections:
In the section Materials and Methods (line 76)
- Since 25OHD levels can be significantly altered due to different exposure to UVB, in a European GWAS meta-analysis, Manousaki et al, selected “winter” individuals assessed January-March (n = 98,674) and “summer” individuals assessed July-September (n = 95,135), while Individuals with vitamin D levels assessed in spring (April-June) and fall (October-December) were not included.
Can you clarify in what season the vitamin D concentration measurements were performed in subjects included in this study?
- Where there any vitamin D supplement users in your study cohort?
- If yes, did you performed any adjustments of 25OHD levels?
The SNPs used as instruments in this study indeed come from the Manousaki et al 25OHD GWAS (PMID:32059762) the reviewer is referring to. The 25OHD measurements were adjusted for 4 seasons in this GWAS. What the reviewer is referring to is an interaction analysis in the same paper, where only two seasons were retained. In this GWAS, a UKBB GWAS was performed and meta-analyzed with a prior GWAS. While both GWAS adjusted for age, sex and season of 25OHD measurement, only the UKBB GWAS took into consideration vitamin D supplement use. Here are the parts of the aforementioned GWAS publication (PMID: 32059762) explaining this:“Data on 25OHD level (in nmol/L) measured using the Diasorin assay were available from 465,415 samples, representing 449,978 UK Biobank participants. Measurements were performed at baseline (2006–2010) and/or the first follow-up visit (2012– 2013). In the present study, we used baseline 25OHD measurements from 401,460 individuals from the white British subset of UK Biobank, as defined below. To account for vitamin D supplement use, we adjusted 25OHD levels by subtracting 21.2 nmol/L from the 25OHD measurement in 24,874 vitamin D supplement users, representing 6% of our study cohort… We then tested the additive allelic effects of SNPs on 25OHD levels, using a linear mixed-model in the BOLT-LMM software. The model-fitting was performed on hard-called genotypes from 488,377 participants consisting of 803,113 SNPs. Age, sex, season of 25OHD measurement (as a categorical variable; 1 for winter [January to March], 2 for spring [April to June], 3 for summer [July to September], and 4 for fall [October to December]), geno- type batch, genotype array, and assessment center (as a proxy for latitude) were included as covariates in the BOLT-LMM.”
As such, in our MR analysis, we did not perform any adjustments of the effect of the SNPs. We used betas and standard errors derived directly from the GWAS. We now specify in the text that “25OHD levels were adjusted for age, sex, season of 25OHD measurement in the entire GWAS and additionally for vitamin D supplementation in the UK Biobank subset of the GWAS meta-analysis”.
Besides limitations which are stated by authors, I would add one more, which refers to the fact that all individuals were from the same or similar geographical area. Even though mixed-ancestry was one of the focus of examination, there is no geographical diversity among them.
We have added the following sentence in the limitations of this study: “Another limitation is the fact that both the 25OHD and youth-onset T2D GWAS included participants from restricted geographical areas, and as such the findings need to be validated in ethnic -diverse populations with a more various geographic distribution.”